# A Review of Carbon Dots Produced from Biomass Wastes

**DOI:** 10.3390/nano10112316

**Published:** 2020-11-23

**Authors:** Chao Kang, Ying Huang, Hui Yang, Xiu Fang Yan, Zeng Ping Chen

**Affiliations:** 1School of Chemistry and Chemical Engineering, Guizhou University, Guiyang 550025, China; ckang@gzu.edu.cn; 2Key Laboratory of Tobacco Quality Research of Guizhou Province, College of Tobacco Science, Guizhou University, Guiyang 550025, China; yhuang@gzu.edu.cn; 3Guizhou Academy of Tobacco Science, Guiyang 550081, China; yangh@gzyks.cn; 4State Key Laboratory of Chemo/Biosensing and Chemometrics, College of Chemistry and Chemical Engineering, Hunan University, Changsha 410082, China

**Keywords:** carbon dots, biomass wastes, synthesis method, recycling, review

## Abstract

The fluorescent carbon dot is a novel type of carbon nanomaterial. In comparison with semiconductor quantum dots and fluorescence organic agents, it possesses significant advantages such as excellent photostability and biocompatibility, low cytotoxicity and easy surface functionalization, which endow it a wide application prospect in fields of bioimaging, chemical sensing, environmental monitoring, disease diagnosis and photocatalysis as well. Biomass waste is a good choice for the production of carbon dots owing to its abundance, wide availability, eco-friendly nature and a source of low cost renewable raw materials such as cellulose, hemicellulose, lignin, carbohydrates and proteins, etc. This paper reviews the main sources of biomass waste, the feasibility and superiority of adopting biomass waste as a carbon source for the synthesis of carbon dots, the synthetic approaches of carbon dots from biomass waste and their applications. The advantages and deficiencies of carbon dots from biomass waste and the major influencing factors on their photoluminescence characteristics are summarized and discussed. The challenges and perspectives in the synthesis of carbon dots from biomass wastes are also briefly outlined.

## 1. Introduction

First discovered by Xu et al. in 2004 [1] and subsequently named by Sun et al. in 2006 [2], carbon dots (C-dots) are novel fluorescent carbon nanomaterials and among the most important members of the carbon nanomaterials family. In general, C-dots are well-dispersed spherical particles with particle size less than 10 nm. Besides the high quantum yield and adjustable emission wavelength, which are also possessed by traditional semiconductor quantum dots, C-dots have many other excellent characteristics, including good photostability, low cytotoxicity, good biocompatibility, easy surface modification and high chemical inertness, and therefore have attracted considerable scholarly attention in recent years. Hence far, C-dots have been widely used in many fields such as cell imaging [3,4,5], in vivo imaging [6,7], drug delivery [8,9,10], fluorescence sensing [11,12,13], photocatalysis [14,15,16], multicolor light-emitting diode (LED) production [17,18], energy conversion and storage [19,20,21], etc. C-dots are gradually become one of the research hotspots in the above-mentioned fields and considered as a potential substitute for semiconductor quantum dots.

The controllable synthesis of carbon dots is currently still in the early stage of its development. The synthesis methods can be divided into two main categories, top-down and bottom-up approaches (Figure 1). The top-down approaches include electrochemical oxidation [22,23,24], arc discharge [25] and laser ablation [26,27,28], which involve the exfoliation process of larger carbonaceous materials (e.g., large-size graphene, carbon nanotube, graphite, commercial activated carbon) into nanoscale C-dots. During the synthesis of C-dots using top-down approaches, harsh experimental conditions (e.g., strong acid and arc discharge), tedious operation steps and expensive equipment are usually employed, which greatly limit their practical application. The bottom-up approaches, such as the microwave-assisted method [29,30,31], pyrolysis [32,33,34], solvothermal method [35,36,37], ultrasonic method [38,39,40], on the contrary, convert small molecules into C-dots via carbonization and passivation. They have the advantages of cost-effectiveness, easy operation and simple equipment requirements, and hence have been widely used in the synthesis of C-dots.

The raw materials for the synthesis of C-dots are very abundant and can be classified into two types, i.e., organic and inorganic carbon sources. The fluorescent quantum yields of C-dots produced from inorganic carbon sources are relatively low [22,41]. Further surface passivation of bare carbon dots is necessary to enhance their luminescent efficiency [42]. In practice, organic carbon source such as organic compounds, organic natural products and biomass waste is more popularly used in the preparation of C-dots [43,44,45].

## 2. Synthesis of C-dots from Biomass Wastes

Biomass is a complex, abundant, heterogeneous, biodegradable and bio-organic substance that may be obtained from diverse sources such as perennial grass, organic domestic garbage, residues of agriculture, fishery, poultry, animal husbandry, forestry and related industries [46,47,48] (Figure 2). Biomass waste is a natural organic carbon source, mainly composed of cellulose, hemicellulose, lignin, ash, proteins and some other ingredients. Taking plant biomass waste as an example, among which, cellulose accounts for 30–60%, hemicellulose 20–40%, lignin 15–25% [49]. Biomass waste is a renewable, environmentally friendly, abundantly available and innocuous carbon source for C-dots production. However, most of the biomass waste is currently discarded, landfilled or openly burned, which not only lead to a waste of resource but also cause some environmental problems threatening human health [50]. Recently, there have been some attempts to utilize biomass wastes as raw materials in the production of C-dots. In the following sections, the methods for the synthesis of carbon dots from biomass waste are discussed.

### 2.1. Pyrolysis

Pyrolysis is a widely used approach for preparing C-dots. The organic substance in the carbon source is gradually converted into carbon dots via heating, dehydration, degradation, carbonization under high temperature in either vacuum or inert atmospheres. High-concentration acid or alkali is generally used in the pyrolysis method to cleave carbon precursors into nanoparticles. All kinds of biomass materials, including watermelon peel, sago waste, coffee grounds and plant leaves, could be used as carbon sources for producing C-dots by pyrolysis method. The properties of the obtained C-dots can be regulated by changing the conditions of pyrolysis, such as pyrolysis temperature, pyrolysis duration, and the pH value of the reaction systems [51]. Zhou et al. [52] achieved large-scale production of C-dots by pyrolysis of waste watermelon peels under low-temperature followed by filtration. The obtained C-dots possess strong blue luminescence, excellent water solubility, good stability in solutions with a wide range of pH and high salinity. The as-prepared carbon dots were successfully used in HeLa cell imaging (Figure 3a). Fluorescence C-dots with a quantum yield of 10.6% and low inherent cytotoxicity was obtained by pyrolysis from lychee seeds [53] and used for fluorescence imaging of living HepG2 cells (Figure 3b). Xue et al. [54] proposed a simple and economical pyrolysis method for the synthesis of fluorescent C-dots from peanut shell waste. The C-dots obtained were N-doped carbon nanomaterials with a quantum yield of 9.91% and possessed good stability, good photo-bleaching resistance, high tolerance to large fluctuation in pH and ionic strength. The surface of the N-doped carbon dots contained a variety of functional groups such as hydroxyl, carboxyl and amino groups. The authors claimed that the emission spectrum of the C-dots prepared by their method was excitation-wavelength-dependent, which was attributed to a wide range of particle sizes and emissive sites. This kind of C-dot was successfully applied to live-cell multicolor imaging (Figure 3c). Praneerad and coworkers [55] prepared C-dots from durian peel waste by pyrolysis method and used them as an effective dopant to construct a composite electrode with a specific capacitance of 60 F·g-1, much higher than that of the pure activated carbon electrode. Sago industrial waste was also used as a carbon source for the synthesis of C-dots by a simple thermal pyrolysis method [56]. The effects of pyrolysis conditions on the physical appearance, particle sizes and fluorescence intensities of C-dots produced from sago industrial waste were investigated in detail [56]. In another article, bio-waste sago bark was also used as a starting precursor for the synthesis of C-dots through a simple one-step pyrolysis method [57], the results obtained by cell viability assay suggested that the C-dots produced from sago industrial waste have low cytotoxicity to cells, and hence can be potentially used for cell imaging. Furthermore, the porosity of the C-dots obtained by pyrolysis approaches endows them with potential applications in anticancer drug delivery.

### 2.2. Solvothermal Method

The solvothermal method is quite popular in the synthesis of C-dots. The organic or inorganic carbon source is mixed with one or several solvents in a stainless steel autoclave with a Teflon liner. After a certain period time of continuous heating, the raw carbon source will be converted to C-dots in the air or inert atmosphere under high temperature and pressure conditions. The solvothermal method has the advantages of environmental friendliness, low cost, easy operation and requirement of simple equipment. A variety of carbon precursors can be used for the synthesis of C-dots by the solvothermal method. Table 1 summarized the properties and applications of C-dots synthesized from various biomass waste by solvothermal method. For example, Lu et al. [58] utilized pomelo peel to synthesize water-soluble C-dots with a quantum yield of 6.9% for the selective and sensitive determination of Hg^2+^ by hydrothermal method (Figure 4a). Prasannan et al. [59] proposed a facile one-pot hydrothermal carbonization method for the synthesis of fluorescent C-dots from orange waste peels under mild conditions and composited the C-dots with ZnO for photo-catalyzing degradation of naphthol blue-black azo dye under UV irradiation (Figure 4b). Tyagi et al. [60] obtained C-dots with spherical morphology and oxygen-rich surface functionalities from lemon peel waste through a facile hydrothermal process and applied them to the determination of Cr^6+^ and the preparation of TiO_2_–C-dots for the photocatalytic degradation of methylene blue dye under UV light irradiation (Figure 4c). Liu et al. [61] synthesized C-dots with a quantum yield of 7.1% from waste bamboo leaves via the hydrothermal method and coated the C-dots with branched polyethylenimine through electrostatic adsorption for the selective and sensitive detection of Cu^2+^ in river water (Figure 4d). Moreover, Sarswat et al. [62] utilized food, beverage and combustion wastes as resources to produce multicolor and highly luminescent C-dots for the fabrication of light-emitting diodes. In Table 1, carbon dots obtained from various biomass waste by the solvothermal method were demonstrated. Obviously, due to the high efficiency and easy operation, the solvothermal method is one of the most frequently used methods for carbon dots preparation.

### 2.3. Microwave-Assisted Method

The microwave-assisted method is a widely employed method for directly carbonizing organic substances into C-dots under microwave radiation. Due to its efficiency, simplicity in terms of device and operation, the microwave-assisted method is a cost-effective approach with a strong competitive edge for producing large amounts of fluorescent C-dots. Wang et al. [78] proposed a waste-reused and eco-friendly microwave-assisted approach for preparing C-dots from protein-rich kitchen waste of eggshell membranes. The obtained C-dots possessed good water solubility and excellent fluorescent with a quantum yield of about 14% and were used for the simultaneous determination of Cu^2+^ and glutathione (Figure 5a). Kumawat et al. [79] developed a relatively simple one-pot microwave-assisted method for preparing C-dots from *Mangifera indica* leaves. Their C-dots exhibited excitation-independent NIR emission, superior cellular uptake, high photostability, good biocompatibility and intracellular temperature sensing capability. (Figure 5b). A one-pot microwave-assisted hydrothermal method was reported by Yao et al. [80] for the synthesis of novel magnetofluorescent carbon quantum dots (i.e., Gd@CQDs, Mn@CQDs and Eu@CQDs, respectively) from waste crab shell and transition-metal ions Gd^3+^, Mn^2+^ and Eu^3+^ (Figure 5c). The obtained magneto-fluorescent C-dots exhibited high stability in a wide range of pH and NaCl concentration. Among the three C-dots/transition-metal ions composites, Gd@CQDs can conjugate with folic acid through a covalent bond to form Gd@CQDs, which exhibits specific targeting property to cancer cells with overexpressing folate receptor. Therefore, FA-Gd@CQDs could be a promising candidate for drug delivery. Bankoti et al. [81] produced C-dots co-doped with nitrogen, sulfur and phosphorous through simple microwave treatment of culinary waste onion peel. The as-prepared C-dots displayed strong green luminescence, high stability against pH and UV exposure, excellent blood compatibility, good cytocompatibility, free radical scavenging activity and efficacy in accelerating wound healing (Figure 5d).

### 2.4. Ultrasonic-Assisted Method

Ultrasonic-assisted method for the synthesis of C-dots has the advantages of low cost and easy operation. C-dots could be obtained through the ultrasonic treatment of mixtures of solvents and carbon sources. The properties of C-dots could be regulated by simply adjusting the experimental conditions such as the ultrasonic power, reaction time, the proportion of solvents and carbon sources, etc. Park et al. [82] proposed a simple method based on ultrasonic treatment for the large-scale synthesis of water-soluble C-dots from food waste-derived carbon source and achieved the production of 120 g C-dots produced from 100 kg of food waste mixtures (Figure 6). The obtained C-dots exhibited high water solubility, high photostability, good photoluminescence and low cytotoxicity for in vitro bioimaging. Furthermore, the byproducts produced in the synthesis of C-dots from the food-waste-derived source could promote seed germination and plant growth.

### 2.5. Other synthetic Methods

Thambiraj et al. [83] developed a top-down approach involving chemical oxidation and a simple exfoliation process for green synthesis of fluorescent C-dots from waste sugarcane bagasse pulp. The as-prepared C-dots had features of high fluorescent quantum yield (about 18.7%), highly crystallinity and excellent biocompatibility (Figure 7a). Sugarcane bagasse was also used as a carbon source by Jiang et al. [84] in the preparation of red-emitting C-dots for selective determination of gaseous ammonia via carbonization using concentrated sulfuric and phosphoric acid (Figure 7b). One-step synthesis of sulfur-doped C-dots from waste frying oil by the use of concentrated sulfuric acid as the carbonization agent was proposed by Hu et al. [85]. Due to its strong dehydrating and oxidizing properties, concentrated sulfuric acid could promote the conversion of saturated C-C single bonds into unsaturated C=C double bonds and the formation of hydrophilic C-O-H and O=C-O-H from hydrophobic C-H. The C-dots synthesized from frying oil displayed a pH-sensitive fluorescent behavior and therefore were used the fluorescence probe for sensing and imaging pH changes in living cells (Figure 7c). Cheng et al. [86] reported a chemical cutting approach for preparing fluorescent C-dots from walnut shells. The prepared C-dots were featured with good photostability, resistance to high ionic strength, stable up-conversion fluorescence and excellent cytocompatibility, which could be applied to the imaging of living cells and tissues (Figure 7d). C-dots with high fluorescence quantum yield of 36% were prepared from polystyrene foam waste by Zhang et al. [87] using a facile synthesis method based on the traditional organic solvent extraction. C-dots with different emission wavelengths were obtained under different solvent conditions. The authors studied the mechanisms of the formation process and the luminescence performance of the C-dots. The effectiveness of surface passivation was considered as the main factor causing the difference in fluorescence emission wavelengths of C-dots.

Among the above-mentioned synthetic approaches for the preparation of C-dots, the hydrothermal approach is the most widely used one for its advantages of convenience, cost-effectiveness, easy operation and eco-friendliness. However, the synthesis of C-dots by the hydrothermal approach is generally considered a time-consuming process. Although the microwave-assisted method is not used as frequently as hydrothermal and pyrolysis approaches, its features of time-saving, low cost and easy operation are extremely attractive for the green synthesis of C-dots from renewable biomass wastes.

## 3. Major Factors Affecting the Properties of C-dots

Quantum size effects, surface defect states, bandgap transition and surface passivation are recognized as the four major factors governing the fluorescence properties of C-dots [88,89,90]. There is research suggesting that surface state is one of the most important mechanisms for explaining the photoluminescence properties of C-dots [91]. Surface states of C-dots are mainly induced by surface oxidation of C-dots, which can cause defects on the surfaces of C-dots and hence alter their luminescence. As the degree of surface oxidation of C-dots increases, more surface defects are formed to trap more excitons, resulting in a red-shift of the emission wavelength of C-dots. Functional groups on the surface of C-dots can significantly affect the fluorescence properties of C-dots. Therefore, surface passivation or surface functionalization plays a vital role in regulating the luminous properties of C-dots. Surface passivation treatment of C-dots can enhance photoluminescence intensity, alter emission wavelength [87], narrow fluorescence peak’s width and improve water dispersibility [92]. Furthermore, some other properties of C-dots, such as selective aggregation, could also be regulated to some extent by surface passivation. For example, improved cellular selectivity for nucleoli staining was achieved by simple amine passivation of C-dots synthesized from cow manure [93]. It has been revealed that the luminescence behavior of C-dots is governed by the combination of two or more of the four factors rather than any one of them alone [62]. Since alteration of the synthesis conditions is the primary means of regulating the properties of C-dots, the impact of some of the major synthesis conditions on the properties of C-dots is therefore discussed in the following.

### 3.1. The Impact of Raw Materials

The raw materials utilized in the synthesis of C-dots could affect the fluorescence properties of C-dots. For example, obvious differences in some important properties of the C-dots obtained by the same synthetic approaches from cucumber peels and pineapple peels were observed [94]. After a few weeks of storage, the C-dots obtained from pineapple peels were fully degraded, while the C-dots from cucumber peels showed good stability. Moreover, fungal formed on the surfaces of the C-dots from pineapple, but such a phenomenon was not observed on the C-dots from cucumber peels. Therefore, the C-dots synthesized from cucumber peels have greater application potential in fields of organic electronics and bioimaging. Boruah [95] also found that C-dots prepared from sugarcane bagasse, garlic peels, and taro peels by the same synthetic method had quite different quantum yields (i.e., 4.5%, 13.8% and 26.2%, respectively) and physicochemical properties.

### 3.2. The Effect of Synthesis Temperature

The synthesis of fluorescence C-dots with biomass wastes as carbon sources involves a carbonization process. Since carbonization is an endothermic process, the temperature is a key factor in the synthesis of C-dots. Zhu et al. [96] investigated the influence of pyrolysis temperature on the luminescence properties of C-dots synthesized from various plant leaves (e.g., lotus leaves, pine needles, the oriental plane leaves and palm leaves) via a simple and low-cost pyrolysis method. Their experimental results suggested that the optimum pyrolysis temperatures were different when different carbon sources were used to synthesize C-dots (Figure 8). The pyrolysis temperature should not be too low or too high. A low pyrolysis temperature may prevent the complete carbonization of the carbon source to C-dots. However, at a very high temperature, the carbon source will be over-oxidized, and the surface structure of C-dots will be destroyed, thus causing deterioration in the optical performance of the C-dots. The influence of pyrolysis temperature on the properties of C-dots synthesized from sago waste was studied in detail by Tan et al. [56]. It was found that the particle size of C-dots decreased as the pyrolysis temperature increased. Moreover, the maximum emission wavelength of the C-dots showed a gradual blue-shift. Moreover, pyrolysis temperature also affected the physical appearances of the end-products. The effect of temperature upon C-dots prepared by the hydrothermal approach was similar to that of the pyrolysis method [85].

### 3.3. The Effect of Reaction Time

The effect of reaction time on the optical properties of C-dots is quite similar to that of reaction temperature [96]. A long reaction time will lead to the destruction of the surface structure of C-dots due to over-carbonization. While a short reaction time will cause insufficient carbonization of carbon source, and hence result in C-dots with the weak fluorescence emission. It should be noted that the effect of reaction time on the optical properties of C-dots is temperature-dependent. The optimization of reaction time is meaningful only when the reaction is carried out under a proper temperature. If the reaction temperature is not high enough, no useful end-product will be obtained even when ultra-long reaction time is adopted [85]. Devi et al. [97] optimized the reaction time for the synthesis of C-dots by an environmentally friendly pyrolysis method from a major waste byproduct of the dairy and cheese industry. C-dots with fluorescence quantum yield of 11.4% were obtained under the conditions of pyrolysis time of 30 min and pyrolysis temperature of 220 °C and successfully applied to the detection of selenite in water. Further increase in reaction time could lead to a significant decrease in the fluorescence emission intensity of the end-product. It was also observed by Bandi et al. [65] that reaction time had a significant impact on the fluorescence quantum yield (QY) of C-dots synthesized from onion waste by a simple hydrothermal approach. QY of C-dots first increased with the increase of reaction time and then decreased along with further prolonging of reaction time. A similar relationship between the reaction time and QY of C-dots synthesized from waste rice husk was reported by Ngu et al. [98].

### 3.4. The Effect of pH Value

For C-dots obtained from different carbon sources by different synthetic approaches, the influence of the pH value of the solution on their fluorescence emission intensities varies. Some C-dots prefer neutral pH [53,58,68,74,99] some perform well under acidic pH Waste frying oil as, some work properly only in an alkaline environment [73,85,99], and others are stable over a wide range of pH values [52,80]. For C-dots derived from some biomass waste, their fluorescence emission intensities change in response to a change in pH value, and hence they can be used as pH sensors. For example, the C-dots synthesized from coconut husk by Chunduri et al. [76] using a hydrothermal method possessed plenty of carboxyl and hydroxyl groups on their surface. Due to the protonation and deprotonation of carboxyl groups under different pH conditions and the resulting change in electrostatic charging property, the fluorescence emission intensity of the C-dots decreased gradually as the pH value increased from 4 to 12.

### 3.5. The Effect of Heteroatom Co-Doping

Heteroatom-doping has been used by many researchers to improve the properties of C-dots [100,101,102]. The introduction of heteroatoms into the carbon atom framework can effectively regulate the electrical properties, internal and surficial chemical properties of C-dots. Due to its similar atomic size with a carbon atom, five outermost orbital electrons and higher electronegativity than carbon, the nitrogen atom is usually doped into carbon materials [103]. Moreover, Sulfur atom-doping can regulate the energy state density of C-dots and hence enhance the fluorescence emission intensity of C-dots [103]. Nitrogen and sulfur co-doped C-dots were synthesized from green tea leaf residue by Hu et al. [104] using an approach based on pyrolyzation at high temperature and oxidation in concentrated H_2_SO_4_. The fluorescence quantum yield of the as-prepared C-dots was as high as 14.8%, more than 3 times higher than that of the undoped C-dots.

### 3.6. The Effect of Surface Passivation

The fluorescence quantum yield of bare C-dots without modification is, in general, relatively low. Surface passivation or functionalization is often used to improve the fluorescence emission intensity of C-dots and hence extend their applicability to bioanalytical assays. Surface passivation of C-dots can decrease the surface defects and increase exciton-hole recombination probability, which will prevent C-dots from agglomeration and enhance their fluorescence emission intensity. Surface passivation includes two major approaches, i.e., capping bare C-dots with some long-chain agents and oxidizing the surfaces of C-dots with strong acids. Due to the usual presence of long-chain carboxylic acids and some other functional substances in carbon sources, carbonization and passivation occur at the same time during the synthetic process of C-dots from biomass waste. As a result, C-dots synthesized from biomass waste are often self-passivated. Their surfaces are enriched with hydroxyl, amine, carboxyl or thiol groups [105]. Self-passivated C-dots prepared from biomass waste, therefore, generally exhibit good optical properties.

## 4. Properties of C-dots Obtained from Biomass Waste

C-dots have the advantages of easy operation, low cost, excellent waster-solubility, outstanding photostability, stable photoluminescent, low cytotoxicity, good biocompatibility and easy surface functionalization. Some of these properties will be discussed in the following:

### 4.1. Structural Property

C-dots are generally three-dimensional clusters with a spherical-like structure composed of carbon atoms and tiny amounts of molecules. The inner parts of the three-dimensional clusters contain mainly SP^3^ hybridization carbon atoms and a small portion of SP^2^ hybridization carbon atoms. The crystal lattices of C-dots are well consistent with those of amorphous carbon and graphite [106,107,108]. C-dots usually have a particle size of less than 10 nm, and therefore exhibit “quantum size effect”. When the particle size of C-dots increases, their maximal fluorescent emission wavelengths show a red-shift. C-dots mainly contain elements of C, H, O and N. The proportions of these elements are different for C-dots prepared by different synthetic methods. Surface passivation and some other treatments enriching the surfaces of C-dots with amino, hydroxyl and carboxyl functional groups can improve the water solubility of C-dots and make the surface functionalization easier, which can lay an important foundation for extensive application of C-dots [109,110,111]. Compared with traditional semiconductor quantum dots or other nanoparticle sensors, C-dots can enter cells more easily through endocytosis due to their smaller particle size, and hence have a wider application in cell imaging and quantification of small molecules in cells [3,101,112].

### 4.2. Optical Properties

#### 4.2.1. UV-Absorption Property

C-dots have broad and strong absorption bands in the ultraviolet to visible wavelength region. For C-dots prepared by different synthetic approaches from different precursors and dispersed in different solvents, their absorption spectra are obviously different from each other. In general, C-dots have one or more absorption peaks located at the ultraviolet to visible wavelength region [64,66,113]. The absorption peak in the wavelength range of 220~270 nm can be roughly ascribed to the π-π* transition of C=C and C=N bonds. The peaks located at the wavelength range of 280~350 nm are corresponding to the n-π* transition of C-O and C=O bonds. Absorption peaks lying in the region of 350~600 nm are generally related to the transition of functional groups on the surfaces of C-dots [87]. Moreover, the degree of surface oxidation of C-dots can, to a large extent, affect the location of their absorption peaks [66].

#### 4.2.2. Fluorescence Property

The fluorescence property is one of the most important features of C-dots, which affects their application in many fields. C-dots possess some excellent fluorescence properties, including wide excitation spectrum, narrow emission spectrum, size-dependent (or excitation wavelength-dependent) fluorescence emission, good fluorescence stability, up-conversion luminescence and strong resistance to photobleaching. Though quantum confinement effect, emissive traps, the exciton of carbon, passivated surface defects, oxygen-containing groups or aromatic structures, and free zigzag were used to explain the luminescence of C-dots [87,114,115,116], the exact luminescence mechanism of C-dots has not yet been clearly understood. Zhang et al. [87] found that the fluorescence emission peak position of C-dots synthesized from polystyrene foam waste was affected by the type of organic solvent used to extract the C-dots. It was speculated that surface passivation by different organic solvents formed different surface defects on the surface of C-dots and introduced different emission sites upon C-dots, leading to the changes in the peak position and intensities of the fluorescence emission spectrum of C-dots. Ding et al. [117] prepared a series of C-dots by a hydrothermal method and separated them for the mother liquor by silica column chromatography. Though these C-dots had similar distributions of particle size and graphite structure in their carbon cores, their fluorescence emission peaks positions varied from 440 to 625 nm (Figure 9). The emission wavelength was red-shifted as the degrees of surface oxidation increased. Hence, it was concluded that the degree of surface oxidation was a dominant factor affecting the fluorescence emission of C-dots. For heteroatom-doped C-dots, the binding configuration between heteroatoms and carbon atoms is the key parameter determining the fluorescence properties of the C-dots.

#### 4.2.3. Up-Conversion Fluorescence Property

Instead of traditional down-conversion fluorescence, some C-dots obtained from biomass waste show special upconversion fluorescence with emission wavelength shorter than the excitation wavelength [86,117,119]. Two kinds of popular mechanisms of anti-stokes photoluminescence and multiphoton active process are usually used to explain the phenomenon of upconversion fluorescence. Wu’s group [87] prepared C-dots with upconversion photoluminescence properties from walnut shells and introduced an electronic transition process model to explain the upconversion phenomenon. The electronic transition process model says that an increase in the particle size of C-dots causes a decrease in the gap between LOMO and HOMO. The carbine ground-state multiplicity induces the electronic anti-stokes from the energy level of π to σ. The electrons at the π orbitals can be excited by low energy light with wavelengths over 600 nm and transit to the excited state with higher energy (LOMO). When the electrons return back to the ground -σ state in the form of irradiation, upconversion luminescence occurs (Figure 10). Sun et al. [117] suggested that a multiphoton active process may be the most likely mechanism for the upconversion luminescence of sulfur and nitrogen co-doped C-dots obtained from hair fiber (Figure 11). C-dots with upconversion fluorescence property have special advantages in optical in vivo imaging due to the strong tissue penetration of red light and weak auto-fluorescence interference in the red–light region from biological tissues. The upconversion photoluminescence property of C-dots also lays a good foundation for the practical application of C-dots in the field of two-photon imaging.

### 4.3. Cytotoxicity and Biocompatibility

Due to their excellent optical properties mentioned above, C-dots have wide applications in the field of bioanalysis. Moreover, C-dots possess low cytotoxicity and good biocompatibility, which satisfies the basic requirement of in vivo imaging of cells and tissues [120,121,122]. Mewada et al. [121] prepared C-dots from *Trapa bispinosa* peel through refluxing water-extracted solution under 90 °C. Their experimental results suggested that the obtained C-dots had low cytotoxicity and good biocompatibility on Madin–Darby canine kidney (MDCK) cells. Cheng et al. [122] demonstrated that N and S co-doped C-dots (N/S-CDs) prepared from cellulose-based biowaste had high fluorescence quantum yield, low cytotoxicity and good biocompatibility and could be successfully applied to intracellular imaging (Figure 12).

### 4.4. Catalytic Activity

Some C-dots have an excellent photocatalytic activity or the ability to enhance the photocatalytic activity of other catalysts. C-dots synthesized from orange waste peels were loaded on ZnO to produce composite catalyst C-dots/ZnO by Prasannan et al. [59]. Compared with ZnO catalysts, the photocatalytic activity of C-dots/ZnO was enhanced by 15.7% after 45 min irradiation of UV light, suggesting that C-dots played a significant role in the photocatalytic process (Figure 13a). Upon UV irradiation, the electrons from the valence band of ZnO are excited to the conduction band and then transferred to C-dots leaving holes in the valance band of ZnO. C-dots act as excitons reservoir and prevent the recombination of electron–hole pairs in ZnO. The electrons on C-dots can react with oxygen to produce superoxide ions ($O_{2}^{•-}$). The holes on ZnO can react with water to generate hydroxyl radical ($OH^{•}$). Both $O_{2}^{•-}$ and $OH^{•}$ can act as strong oxidation agents for free radical degradation of naphthol blue-black (NBB) azo dye. Gupta’s groups [60] prepared C-dots with spherical morphology and oxygen-rich surface functionalities from lemon peel waste and immobilized them on electrospun TiO_2_ nanofibers to prepare TiO_2_ - water-soluble carbon quantum dots (TiO_2_–wsCQDs) composite, then was used for examining the photocatalytic activity of the as-prepared TiO_2_–wsCQDs composite catalyst. The photocatalytic activity of TiO_2_–wsCQDs composite for the degradation of methylene blue dye (M) was about 2.5 times higher than that of TiO_2_ nanofibers (Figure 13b)

## 5. Applications of C-Dots Obtained from Biomass Wastes

Due to various kinds of excellent properties of small size, strong fluorescence intensity, good photostability, low cytotoxicity, good biocompatibility, easy surface modification, superior stability and excellent water solubility, biomass wastes derived carbon dots have been extensively studied for widely application, such as sensing, in vivo imaging and target therapy, photocatalysis, drug delivery, energy storage, light-emitting diodes and fluorescence ink. For example, owing to the superior fluorescent properties and abundant functional groups on the surface, carbon dots exhibit the great ability of analyte sensing, which is mainly the most widely used field of biomass waste-derived carbon dots. In addition, the properties of good biocompatibility and photostability enable carbon dots as an ideal candidate for bioimaging. Small size, unique optical and electronic properties, good electrochemical conductivity, and active edge site allows carbon dots to be used as energy storage devices. Table 2 lists the main applications of biomass wastes-derived carbon dots.

## 6. Conclusions

Biomass waste is abundant and widely distributed in the natural and living environment. Herein we have reviewed the methods for preparing fluorescent C-dots from biomass waste, analyzed the main factors affecting the fluorescence intensity of C-dots during the synthetic process, and finally discussed the properties and applications of C-dots obtained from biomass waste.

The advantages of synthesizing C-dots from biomass waste include easy availability of carbon source, simplicity in preparation and feasibility of large-scale production. Compared with C-dots produced from chemical agents, some C-dots from biomass waste can be self-passivated during the synthetic process because the abundant presence of carbonaceous compounds in biomass waste enables carbonization and surface passivation to occur simultaneously, and heteroatom-containing compounds or long-chain compounds present in biomass waste can act as surface passivation agents.

C-dots prepared from biomass waste have relatively high fluorescence quantum yield, good photostability, high photocatalytic activity, excellent biocompatibility and low cytotoxicity and therefore have a wide range of promising applications in many areas such as bioimaging, drug delivery, ion sensing and photocatalysis.

## 7. Future Outlook

Although much effort has been devoted to the preparation of multifunctional C-dots from biomass waste, many challenges still exist. First of all, the photoluminescence mechanism of C-dots has not been fully understood. A detailed and widely recognized photoluminescence mechanism of C-dots is highly needed to expand the application of C-dots from renewable biomass waste. Second, it is still a big challenge to realize the large-scale production of high-quality C-dots from biomass waste. The solubility of C-dots in solvents and their potential applications are closely related to the types and extent of surface functional groups upon C-dots. Unfortunately, how to precisely control surface functionalization of C-dots during the synthetic process is also an exasperating problem. The surface of C-dots obtained from biomass wastes are generally functionalized with carboxyl, hydroxyl and amino groups, which endow C-dots with properties of good solubility in water and rapid precipitation in other solvents but may also quench the fluorescence of C-dots. The use of C-dots as fluorescence probes in the quantitative analysis of metal ions or other substances is based on the fluorescence-quenching mechanism. However, there are many factors other than analytes of interest that can quench the fluorescence of C-dots. C-dots with excellent selectivity for the analytes of interest are therefore highly desired. The maximal absorption wavelengths of most C-dots are located in the wavelength range from the ultraviolet to the blue, which limits their application in the field of bioimaging. The preparation of far-red to near-infrared C-dots is becoming a new research hotspot.

## Figures and Tables

**Figure 1 nanomaterials-10-02316-f001:**
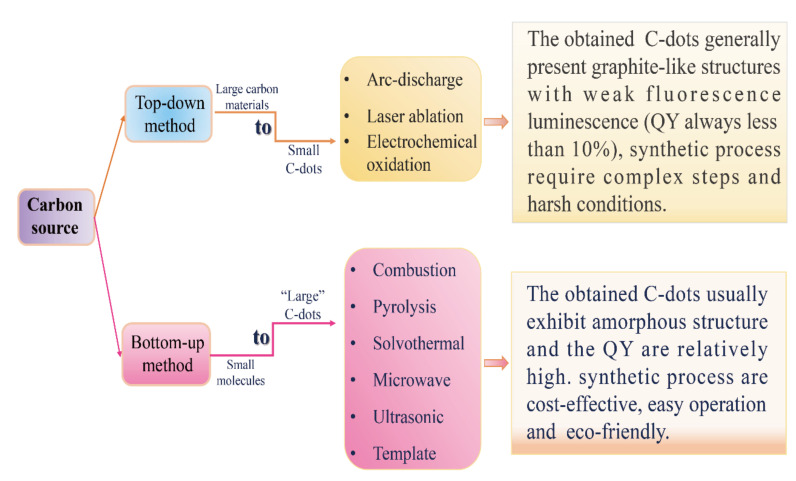
The top-down and bottom-up approaches for synthesizing of C-dots (QY refers to fluorescence quantum yield).

**Figure 2 nanomaterials-10-02316-f002:**
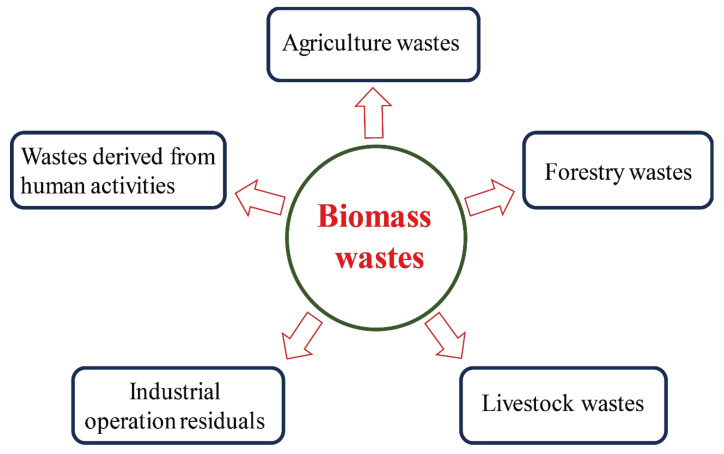
Main sources of biomass waste.

**Figure 3 nanomaterials-10-02316-f003:**
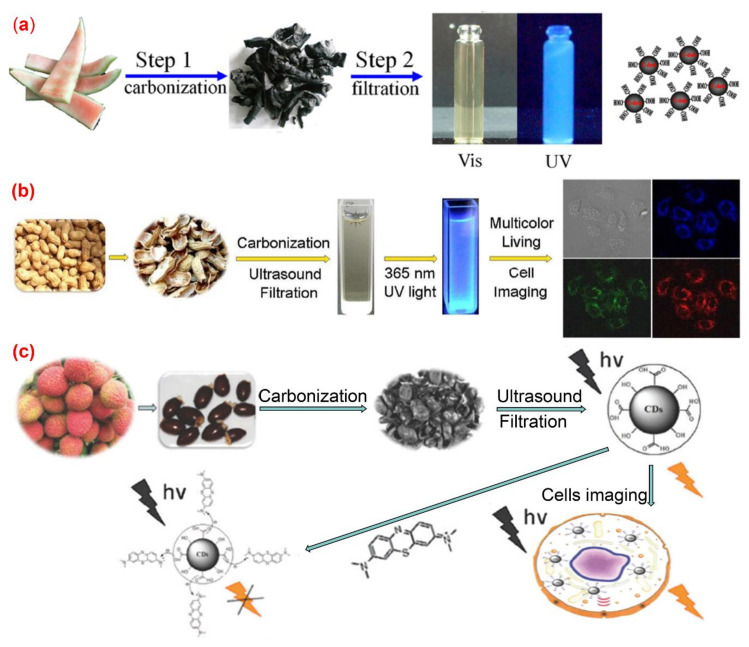
Preparation of fluorescence C-dots from agriculture wastes by pyrolysis method. (**a**) Synthesis of water-soluble C-dots from watermelon peel [52]; (**b**) the preparation and application of fluorescent C-dots from lychee seeds [53]; (**c**) the preparation and application of C-dots from peanut shells [54].

**Figure 4 nanomaterials-10-02316-f004:**
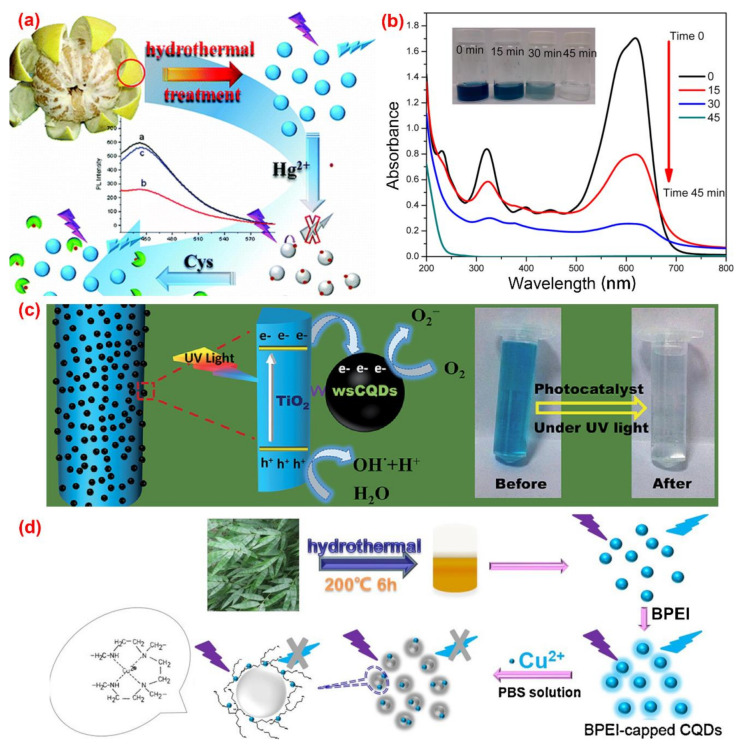
The preparation of fluorescence C-dots from biomass wastes through the hydrothermal method. (**a**) The preparation and application of C-dots from pomelo peel [58]; (**b**) absorption spectra of naphthol blue-black (NBB) azo dye over C-dots/ZnO at different irradiation intervals [59]; (**c**) the photocatalytic degradation of methylene blue (MB) on TiO_2_–C-dots composite under UV light irradiation [60]; (**d**) the synthesis and application of C-dots from bamboo leaves [61].

**Figure 5 nanomaterials-10-02316-f005:**
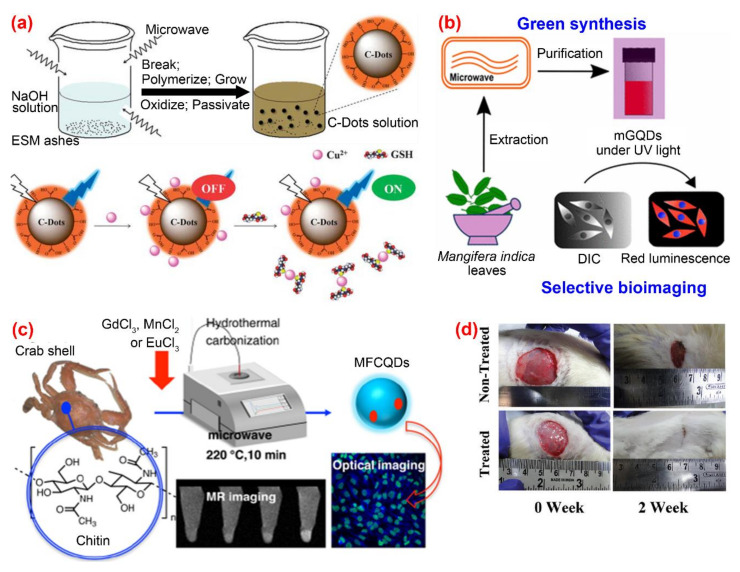
Preparation and application of fluorescence C-dots from biomass wastes by microwave-assisted approaches. (**a**) The preparation and application of C-dots from eggshell membranes [79]; (**b**) the synthesis and application of C-dots from mango leaves [80]; (**c**) the preparation and application of C-dots from crab shell [81]; (**d**) application of C-dots synthesized from onion peel to test the efficacy in accelerating wound healing [82].

**Figure 6 nanomaterials-10-02316-f006:**
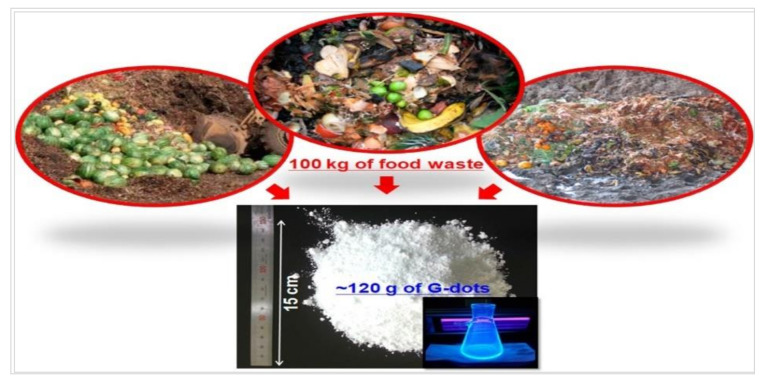
Large-scale synthesis of green C-dots from food wastes [83].

**Figure 7 nanomaterials-10-02316-f007:**
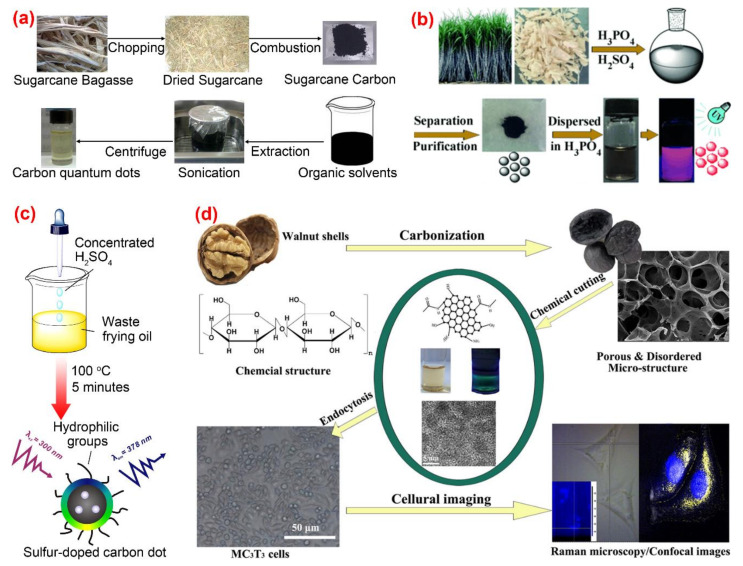
The preparation and application of C-dots by other synthetic methods. (**a**) The synthetic process of C-dots from sugarcane bagasse pulp [84]; (**b**) the synthesis of C-dots from bagasse [85]; (**c**) sulfur-doped carbon dots obtained from waste frying oil [86]; (**d**) the synthesis and application of C-dots from walnut shells [87].

**Figure 8 nanomaterials-10-02316-f008:**
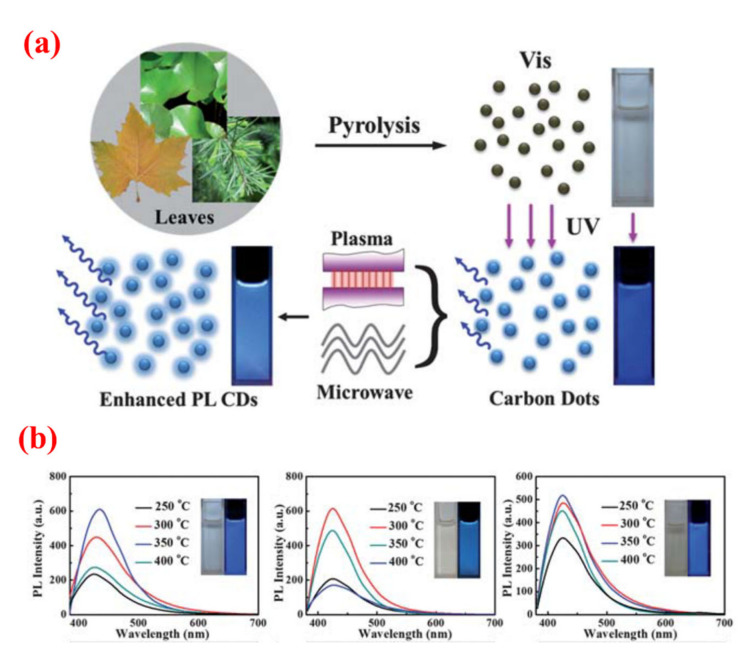
(**a**). Schematic illustration of the synthetic process of C-dots from different plant leaves by the pyrolysis method [97]; (**b**). The fluorescence emission spectra of the corresponding C-dots (obtained from oriental plane leaves, lotus leaves and pine needles successively from left to right) synthesized under different pyrolysis temperatures [97].

**Figure 9 nanomaterials-10-02316-f009:**
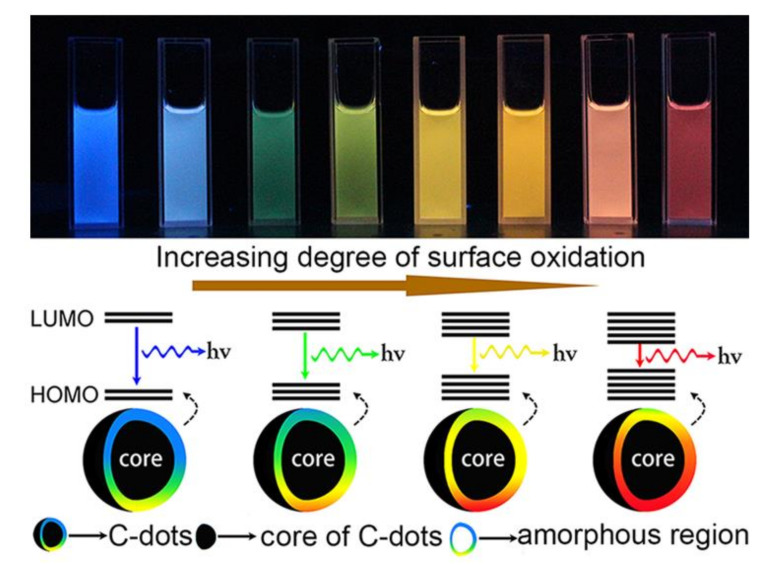
Mechanism of the tunable fluorescent luminescence of C-dots with different degrees of surface oxidation [118].

**Figure 10 nanomaterials-10-02316-f010:**
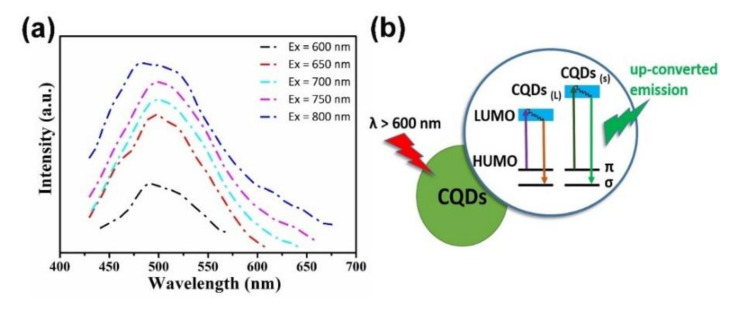
(**a**). Up-conversion photoluminescence properties of C-dots from walnut shells, (**b**). the upconversion processes in C-dots of different particle sizes [86].

**Figure 11 nanomaterials-10-02316-f011:**
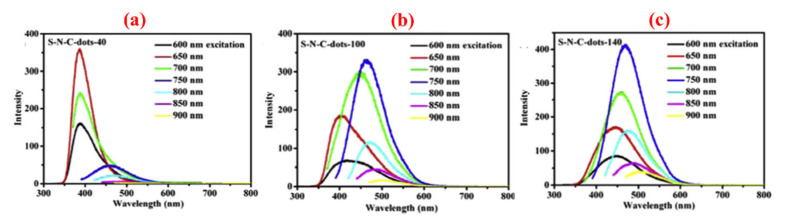
Upconversion photoluminescence properties of S-N-co-doped C-dots obtained from hair fiber under different conditions. (**a**) 40 °C; (**b**) 100 °C; (**c**) 140 °C [117].

**Figure 12 nanomaterials-10-02316-f012:**
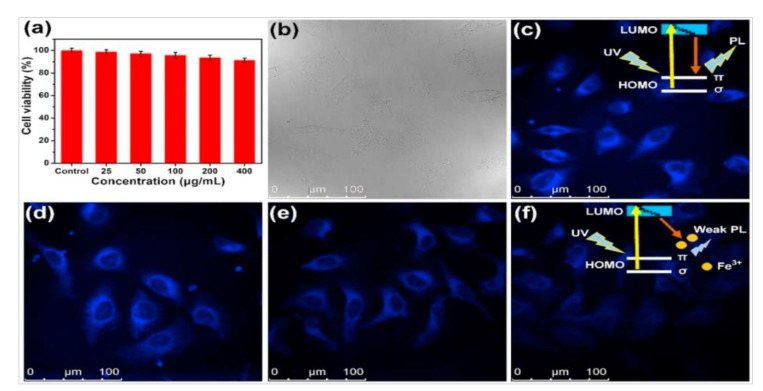
(**a**) Cell viability of HeLa cells after incubated with different concentrations of N/S-doped C-dots obtained from cellulose-based biowaste for 24 h; (**b**–**f**) fluorescence images of HeLa cells incubated with N/S-doped C-dots (40 μg/mL) under different conditions ((**c**): N/S-CDs only, (**d**–**f**): mixture of N/S-CDs and Fe^3+^ with concentrations of 10, 20 and 40 μM, respectively) [122].

**Figure 13 nanomaterials-10-02316-f013:**
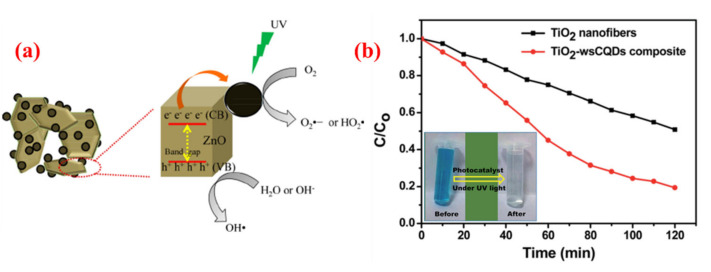
(**a**). Schematic presentation of the photocatalytic process of C-dots/ZnO composite [59]; (**b**). Photocatalytic degradation of MB in the presence of TiO_2_ nanofibers and TiO_2_–wsCQDs composite under UV light irradiation (The inserted picture shows the color of MB solution before and after degradation) [60].

**Table 1 nanomaterials-10-02316-t001:** The properties and applications of C-dots produced from different biomass waste by solvothermal method.

Biomass Waste	Hydrothermal Condition	Fluorescence Quantum Yield	Application	Ref. ^1^
Wheat bran	180 °C, 3 h	-	drug delivery	[63]
Sugarcane Bagasse char	190 °C, 24 h	-	drug delivery	[64]
Waste food	200 °C, 1.5 h	-	light emitting diodes	[62]
Orange peels	180 °C, 12 h	-	photocatalysis	[59]
Onion waste	120 °C, 2 h	28%	Fe^3+^ detection and multicolor imaging	[65]
Waste food	195 °C, 225 °C, 255 °C, 12 h	28%, 18%, 10%, 6% for blue, green, yellow and red C-dots, respectively	Fe^3+^ detection	[66]
Tobacco leaves	200 °C, 3 h	27.9%	three kinds of tetracyclines detection	[67]
Coffee grounds	200 °C, 6–10 h	24%	Fe^3+^, Cu^2+^detection	[68]
Rice residue	200 °C, 12 h	23.48%	Fe^3+^ ions and tetracyclines detection	[69]
Bael leaves	170 °C, 5 h	22%	Fe^3+^ detection	[70]
Wheat straw	250 °C, 10 h	20%	labeling, imaging and sensing	[71]
Lemon peels	200 °C, 12 h.	14%	sensing and photocatalysis	[60]
Wheat straw and bamboo residues	180 °C, 4 h	13%	cell imaging and in vivo bioimaging	[72]
Dried lemon peels	200 °C, 6 h	11%	carmine detection	[73]
Tulsi leaves	180 °C, 4 h	9.3%	Pb^2+^ detection	[74]
Magnolia flower	200 °C, 8 h	8.13%	Fe^3+^ detection	[75]
Bamboo leaves	200 °C, 6 h	7.1%	Cu^2+^detection	[61]
Pomelo peels	200 °C, 3 h.	6.9%	Hg^2+^detection	[58]
Coconut husks	200 °C, 3 h	-	pH sensor	[76]
Prawn shells	180 °C, 12 h	-	nitrite detection	[77]

^1^ refers to the reference.

**Table 2 nanomaterials-10-02316-t002:** An overview of many application fields of carbon dots obtained from biomass wastes.

Application Field	Biomass Waste	Method	Application	Ref. ^1^
Sensing	Bagasse waste	hydrothermal	Hg^2+^ detection	[123]
	Crown daisy leaf waste	hydrothermal	Cu^2+^ detection	[124]
	Lignocellulos waste	hydrothermal	Cu^2+^ detection	[125]
	*Sargassum fluitans*	hydrothermal	DNA detection	[126]
	Mango peels	hydrothermal	mesotrione detection	[127]
	Palm shell waste	ultrasonic	nitrophenol detection	[128]
	Waste tea residue	chemical oxidation	tetracycline detection	[129]
	Waste candle soot	chemical oxidation	Hg^2+^ and Fe^3+^ detection	[130]
	Kerosene fuel soot	chemical oxidation	picric acid, Fe^3+^ and Cu^2+^ detection	[131]
Imaging	Onion waste	hydrothermal	multicolor imaging and Fe^3+^ detection	[65]
	Wheat straw and bamboo residues	hydrothermal	cell imaging and in vivo bioimaging	[72]
	Banana peel waste	hydrothermal	in vivo bioimaging	[132]
	Lychee waste	Solvothermal	multicolor cell imaging and Fe^3+^ detection	[133]
	Roasted gram shells	microwave	in vitro cell imaging	[134]
	Food-waste	ultrasonic	in vitro bioimaging	[82]
	Cow manure	chemical oxidation	live-cell imaging with subcellular selectivity	[93]
	Walnut shells	carbonization and chemical cutting	intracellular bioimaging	[86]
	*T. bispinosa* peel	refluxing	cellular imaging	[121]
Drug delivery	Wheat bran	hydrothermal	drug delivery	[63]
	Sugarcane bagasse	burn and hydrothermal	drug delivery vehicle for acetaminophen	[64]
	Waste sago bark	catalyst-free pyrolysis	anticancer drug delivery and cancer cell imaging	[57]
	Crab shells	microwave	drug delivery and targeted dual-modality bioimaging	[80]
	Bamboo leaves	refluxing	drug delivery and tumor imaging	[135]
Photocatalysis	Waste frying oil	hydrothermal	photocatalysis	[136]
	Orange peels	hydrothermal	photocatalysis	[59]
	Lignocellulosic waste	pyrolysis	photocatalysis coupled to pollutant utilization	[137]
	Bitter apple peels	pyrolysis	photocatalysis	[138]
	Lemon peel waste	hydrothermal	photocatalysis and sensing	[60]
Others	Waste food	hydrothermal	Light-emitting diodes	[62]
	Willow leaves	hydrothermal	fluorescent ink and oxygen reduction electrocatalysts	[139]
	Pineapple peels	hydrothermal	electronic security devices and as a memory element	[140]
	Orange waste peels	hydrothermal	nonlinear optical applications	[141]
	Tea and peanut shells	hydrothermal	tea grades discrimination	[142]
	Sugarcane bagasse	refluxed and hydrothermal	naphthalene removal	[143]
	Durian peels	pyrolysis	energy storage device	[55]
	Onion peels	microwave	accelerated skin wound healing and live-cell imaging	[81]
	Waste tea residue	carbonization	used as growth plant stimulator	[144]

^1^ refers to the reference.

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
