# Peer review of "A Review of Carbon Dots Produced from Biomass Wastes"

_nanomaterials, 2020, doi:10.3390/nano10112316_

Round 1

Reviewer 1 Report

Manuscript ID: nanomaterials-995650

 “A review on carbon dots produced from biomass wastes”

Authors:

Chao Kang , Ying Huang , Hui Yang , Xiu Fang Yan * , Zeng Ping Chen *

The manuscript is a review of carbon dots properties obtained from biomass waste. This work is mainly focused on the preparing methods of fluorescent C-dots from biomass waste, describes this methods and reviewed the advantages and applications of C-dots obtained from biomass waste. This paper is well written, but needs some minor revisions.

Some revision suggestions:

  1. In abstract second sentence concerning carbon dots properties should be shortened.
  2. The abbreviation (QY) presented at Figure 1. should be expanded. Authors explain it at page 10 line 292, but (QY) should be also expanded at Figure 1.
  3. Please correct the font in word “organic” see line page 2, line 63.
  4. The brackets in the (Figure 3(a) and (Figure 3(a) are not closed, or it is too much brackets see page 3, lines 88 and 90. The same situation occur in (Figure 4(a) see page 4, lines 125, (Figure 4(b) see page 4, lines 128, (Figure 5(a) see page 6, lines 157, (Figure 5(b) see page 6, lines 160, (Figure 5(a) see page 8, lines 195, (Figure 5(b) see page 8, lines 197 (Figure 14(a) see page 15, lines 436, please correct it.
  5. The abbreviation GSH should be expanded see page 6, line 157.
  6. Figures: 3, 4, 5 and 7 are very low quality, if it is possible please correct it. Some words, inscriptions, chemical formulas as well as pictures are not legible.
  7. Please correct the subscript in the sulfuric acid formula, see page 11, line 321
  8. Please correct the superscript in the SP3 as well as in SP2, see page 12, line 343.
  9. The abbreviation of MDCK cells should be expanded see page 14, line 422.
  10. Please expand the abbreviation in TiO2-wsCQDs as (water soluble carbon quantum dots (wsCQDs), see page 15, line 444.

Author Response

Thank you very much for reviewing our manuscript and giving meaningful advices. Our corresponding responses are listed point by point as follows.

1. In abstract second sentence concerning carbon dots properties should be shortened.

Author Response: Thank you for your suggestion, we have shortened the corresponding sentence in the revised manuscript.

2. The abbreviation (QY) presented at Figure 1. should be expanded. Authors explain it at page 10 line 292, but (QY) should be also expanded at Figure 1.

Author Response: Thank you for your kind reminding, the abbreviation (QY) in Figure 1 has been explained in the revised manuscript (marked in red).

3. Please correct the font in word “organic” see line page 2, line 63.

Author Response: Thank you for your suggestion, the word “organic” has been corrected in the revised paper, please see page 2, line 64.

4. The brackets in the (Figure 3(a) and (Figure 3(a) are not closed, or it is too much brackets see page 3, lines 88 and 90. The same situation occur in (Figure 4(a) see page 4, lines 125, (Figure 4(b) see page 4, lines 128, (Figure 5(a) see page 6, lines 157, (Figure 5(b) see page 6, lines 160, (Figure 5(a) see page 8, lines 195, (Figure 5(b) see page 8, lines 197 (Figure 14(a) see page 15, lines 436, please correct it

Author Response: Thank you for your suggestion, the unclosed brackets have been completed in the revised manuscript.

5. The abbreviation GSH should be expanded see page 6, line 157.

Author Response: Thank you for your suggestion, the abbreviation GSH has been replaced by glutathione, please see page 6, line 184.

6. Figures: 3, 4, 5 and 7 are very low quality, if it is possible please correct it. Some words, inscriptions, chemical formulas as well as pictures are not legible.

Author Response:

Thank you for this helpful suggestion. In the revised manuscript, the quality of these figures, especially the words, inscriptions, and chemical formulas in these figures, have been improved.

7. Please correct the subscript in the sulfuric acid formula, see page 11, line 321

Author Response: Thank you for your reminding, the subscript in the sulfuric acid formula has been corrected in the revised manuscript, please see page 12, line 353.

8. Please correct the superscript in the SP3 as well as in SP2, see page 12, line 343.

Author Response: Thank you for your reminding, the superscript in the SP3 and SP2 have been corrected in the revised manuscript, please see page 12, line 375.

9. The abbreviation of MDCK cells should be expanded see page 14, line 422.

Author Response: Thank you for your suggestion, the full name of MDCK cells has been given in the revised manuscript, please see page 15, line 464.

10. Please expand the abbreviation in TiO2-wsCQDs as (water soluble carbon quantum dots (wsCQDs), see page 15, line 444.

Author Response: Thank you for your reminding, the explanation for the abbreviation of TiO2-wsCQDs has been given in the revised manuscript, please see page 16, line 492.

Reviewer 2 Report

The article summarizes the methods for synthesizing CDs from biomass wastes, properties of CDs and their applications. My comments are listed below.

  1. The manuscript listed various applications of CDs synthesized from biomass wastes. However, it does not include a detail explanation how successful the CDs are. The applications in bioimaging, biosensing, drug delivery, energy devices, etc. can be described in detail giving specific examples from the reported literature.
  2. Is there any difference in using solid biomass and liquid biomass?
  3. Is there any difference in the properties of CDs produced from the biomass and other methods?
  4. According to previously reported papers, the product yield of CDs synthesized using hydrothermal method ranges from very low to high. What is the optimal water-biomass ratio? A range of ratio can be included.

Author Response

Thank you very much for reviewing our manuscript and giving meaningful advices. Our corresponding responses are listed point by point as follows.

1. The manuscript listed various applications of CDs synthesized from biomass wastes. However, it does not include a detail explanation how successful the CDs are. The applications in bioimaging, biosensing, drug delivery, energy devices, etc. can be described in detail giving specific examples from the reported literature.

Author response: Thank you for this helpful advice, the applications of carbon dots from biomass waste are added in the fifth section, we used a table to summarize the main applications of carbon dots produced from biomass wastes, in order to provide a detail explanation how successful the CDs are.

2. Is there any difference in using solid biomass and liquid biomass?

Author response: In the public references, biomass wastes that used for producing carbon dots are solid biomass in general. I think that there exists difference in using solid biomass and liquid biomass, since that different states can lead to different properties (e.g. surface structure and degree of aggregation), this must be an interesting research topic.

3. Is there any difference in the properties of CDs produced from the biomass and other methods?

Author response: Thank you for your meaningful advice, from current researches, there exists no significant difference between carbon dots obtained from biomass wastes and other materials such as chemical regents, nature source of vegetables and fruits. Furthermore, the application of carbon dots is determined by its properties, current researches suggest that the application fields of carbon dots obtained from biomass wastes are consistent with that obtained from other materials, no obvious difference was found among these raw materials.

4. According to previously reported papers, the product yield of CDs synthesized using hydrothermal method ranges from very low to high. What is the optimal water-biomass ratio? A range of ratio can be included.

Author response: Thank you for this helpful comment. Previously reported papers optimized the experimental conditions mostly according to fluorescent quantum yield, the studies on optimal conditions for product yield of CDs are rarely. According to previous references, the range of the water-biomass ratio is generally from 1:100 to 25:100 (Table 1). I think that the study on optimal conditions for product yield of CDs will be an interesting and worthwhile topic in the future.

Table 1. The reaction conditions and corresponding product yield in several reference

Raw materials

Solvent

Mass ratio

Reaction condition

Product yield (%)

Reference

 wheat straw

water

1:50

250℃, 10 h

20

[1]

 prawn shell

water

1:4

180℃, 12 h

18

[2]

orange waste peels

 0.1 M H2SO4

1:50

150℃, 2 h

12

[3]

coconut husk

water

1:50

200℃, 3 h

7

[4]

Onion waste

water

3:10

120℃,2 h

6

[5]

coffee grounds

water

1:100

150–200℃,

6-10 h

33

[6]

[1] Y. Ming, R. Zhong, H. Gao, W. Li, X. Yun, J. Liu, X. Zhao, G. Zhao, Z. Feng. One-step, green and economic synthesis of water-soluble photoluminescent carbon dots by hydrothermal treatment of wheat straw and their bio-applications in labeling, imaging and sensing. Applied Surface Science, 2015, 355:1136-1144.

[2] H. Zhang, S. Kang, G. Wang, Y. Zhang, H. Zhao. Fluorescence determination of nitrite in water using prawn-shell derived nitrogen-doped carbon nanodots as fluorophores. ACS Sensors, 2016, 1, 875-881.

[3] A. Prasannan, T. Imae. One-Pot Synthesis of fluorescent carbon dots from orange waste peels. Industrial & Engineering Chemistry Research, 2013, 52 (44): 15673–15678.

[4] L. A. A. Chunduri, A. Kurdekar, S. Patnaik, B. V. Dev, T. M. Rattan, V. Kamisetti. Carbon quantum dots from coconut husk: evaluation for antioxidant and cytotoxic activity. Materials Focus, 2016, 5 (1): 55-61.

[5] R. Bandi, B. R. Gangapuram, R. Dadigala, R. Eslavath, S. S. Singh, V. Guttena. Facile and green synthesis of fluorescent carbon dots from Onion waste and their potential applications as sensor and multicolour imaging agents. Rsc Advances, 2016, 6 (34): 28633-28639.

[6] W. Liang, W. Li, B. Wu, L. Zhen, S. Wang, L. Yuan, D. Pan, M. Wu. Facile synthesis of fluorescent graphene quantum dots from coffee grounds for bioimaging and sensing. Chemical Engineering Journal, 2016, 300:75-82.
